# AT-101 Enhances the Antitumor Activity of Lenalidomide in Patients with Multiple Myeloma

**DOI:** 10.3390/cancers15020477

**Published:** 2023-01-12

**Authors:** Sikander Ailawadhi, Ricardo D. Parrondo, Navnita Dutta, Bing Han, Gina Ciccio, Yesesri Cherukuri, Victoria R. Alegria, Betsy R. LaPlant, Vivek Roy, Taimur Sher, Brett Edwards, Stephanie Lanier, Alak Manna, Keisha Heslop, Thomas Caulfield, Emir Maldosevic, Peter Storz, Rami Manochakian, Yan Asmann, Asher A. Chanan-Khan, Aneel Paulus

**Affiliations:** 1Deparment of Hematology-Oncology, Mayo Clinic Florida, 4500 San Pablo Road S, Jacksonville, FL 32224, USA; 2Department of Cancer Biology, Mayo Clinic, 4500 San Pablo Road S, Jacksonville, FL 32224, USA; 3Department of Health Sciences Research, Mayo Clinic, Jacksonville, FL 32224, USA; 4Department of Biostatistics, Mayo Clinic Rochester, Rochester, MN 55902, USA; 5Department of Neuroscience, Mayo Clinic, Jacksonville, FL 32224, USA

**Keywords:** multiple myeloma, Bcl-2 inhibition, apoptosis

## Abstract

**Simple Summary:**

Bcl-2 family proteins play a key role in myeloma cell survival and are implicated in drug resistance and development of refractory disease, thus making them an attractive therapeutic target. Currently available strategies targeting Bcl-2 proteins in multiple myeloma have shown a benefit limited to patients with t(11;14) myeloma in the case of venetoclax and studies targeting both Bcl-2 and Mcl-1 have been fraught with cardiovascular toxicity. In this article we present a “bench-to-bedside” evaluation of AT-101, lenalidomide and dexamethasone (ARd). We highlight in this report the in vitro and (mouse model) in vivo efficacy of the ARd regimen and report for the first time, clinical proof of concept, safety and efficacy of ARd in relapsed/refractory multiple myeloma patients through a phase I/II clinical trial.

**Abstract:**

Bcl-2 and Mcl-1 proteins play a role in multiple myeloma (MM) cell survival, for which targeted inhibitors are being developed. AT-101 is an oral drug, which disrupts Bcl-2 and Mcl-1 function, impedes mitochondrial bioenergetic processes and induces apoptosis in MM cells. When combined with lenalidomide and dexamethasone (Rd), AT-101 significantly reduced tumor burden in an in vivo xenograft model of MM. These data provided rationale for a phase I/II study to establish the effective dose of AT-101 in combination with Rd (ARd regimen) in relapsed/refractory MM. A total of 10 patients were enrolled, most with high-risk cytogenetics (80%) and prior stem cell transplant (70%). Three patients were lenalidomide-refractory, 2 were bortezomib-refractory and 3 were daratumumab-refractory. The ARd combination was well tolerated with most common grade 3/4 adverse events being cytopenia’s. The overall response rate was 40% and clinical benefit rate was 90%. The median progression free survival was 14.9 months (95% CI 7.1-NE). Patients responsive to ARd showed a decrease in Bcl-2:Bim or Mcl-1:Noxa protein complexes, increased CD8+ T and NK cells and depletion of T and B-regulatory cells. The ARd regimen demonstrated an acceptable safety profile and promising efficacy in patients with relapsed/refractory MM prompting further investigation in additional patients.

## 1. Introduction

Multiple myeloma (MM) is a plasma cell neoplasm and is the second most common hematologic cancer in the U.S. [1]. Current therapeutic strategies use combinations of immunomodulatory drugs (IMiDs), proteasome inhibitors (PI), and/or monoclonal antibodies (mAbs), which have resulted in improved clinical outcomes for patients with MM [2,3]. However, despite the availability of several therapeutics, MM remains incurable; patients experience frequent disease relapses and ultimately succumb to the disease [4]. Precision therapeutics with the ability to exploit vulnerabilities through targeting pathways that are detrimental to MM cell survival are urgently needed.

MM cells evade death by utilizing a myriad strategies; comprising both cell-intrinsic and cell-extrinsic processes [5]. Upon cell stress (including due to therapy), molecular pathways within the tumor cells are activated to preserve their survival by engaging the Bcl-2 family of proteins. Prior preclinical [6,7,8,9] and clinical studies [10,11] have demonstrated that MM cells overexpress anti-apoptotic Bcl-2 and Mcl-1 proteins, which supports their survival by preserving optimal mitochondrial function. To complement these internal survival mechanisms, unrestrained growth of MM cells is also intricately supported by factors within the tumor microenvironment [12]. MM patients typically have compromised immunity with well-characterized abnormalities in the function of immune effector cells (i.e., CD8+ T-cells or NK-cells) [12]. IMiDs, mAbs and novel cellular therapies (CAR-T) are known to restore host immunity and re-engage anti-tumor immune T or NK cells in MM patients [13].

Recently, we investigated the Bcl-2 selective inhibitor, venetoclax (ABT-199), in relapsed/refractory MM and demonstrated its safety and efficacy both as a single agent [14] and in combination with other active anti-MM agents [15,16,17,18]. An interesting clinical observation with ABT-199 was that favorable response was mainly limited to patients carrying t(11;14) translocation, which only represents ~15% of all MM patients suggesting that myeloma cells in these patients rely predominantly, if not exclusively on Bcl-2 [19]. However, reliance on more than one anti-apoptotic member of the Bcl-2 family may explain the tempered clinical responses with single agent ABT-199 in most MM patients. Thus, agents that target multiple anti-apoptotic proteins may show greater efficacy in MM patients and especially in those without t(11;14).

AT-101 is an oral small molecule inhibitor, with the ability to target both Bcl-2, Mcl-1 and to lesser degree Bcl-xL and Bcl-w [20]. Our preclinical investigation validated that AT-101 by itself can induce apoptosis in various B-cell malignancies including MM [21,22,23]. Given anti-myeloma therapies have now primarily shifted towards multi-agent regimens, we investigated whether activating the intrinsic apoptotic pathway with AT-101 can enhance the cytotoxic potential of existing MM therapeutics. We observed in preclinical models of MM that AT-101, by disrupting Bcl-2 and Mcl-1 function, potentiates the cytotoxic effects of lenalidomide (Revlimid) and dexamethasone (Rd) [21]. In this manuscript we report additional mechanistic evaluation of how the combination of AT-101 and Rd (ARd) undermine MM cell survival in a xenograft model system. Based on these promising studies we report for the first time, clinical proof of concept of the ARd regimen in relapsed/refractory MM patients through a phase I/II clinical trial (NCT02697344) and correlatives analyses’ highlighting the effects of this regimen on Bcl-2 protein dynamics and changes in immune cell populations.

## 2. Materials and Methods

MM cell lines KMS-11, RPMI-8226 and MM1.S were used in experiments. CD138+ tumor cells from patients with a confirmed diagnosis of MM were used after obtaining written informed consent, approved by the Mayo Clinic Jacksonville Institutional Review Board and in accordance with the Declaration of Helsinki.

### 2.1. Seahorse Extracellular Flux Assay

KMS-11 cells (seeded 6 × 10^5^ viable cells in each well, 6 replicates for each group) were treated with 5μM ABT-199, 5μM AT-101, or DMSO vehicle control in RPMI medium with 10% FBS for 24 h in a manner previously described by us.4 The Seahorse XF Cell Mito Stress Test Kit (Agilent 103015-100) was used to measuring cell mitochondrial function using the Agilent Seahorse XF96 analyzer following the manufacturer’s instructions (https://www.agilent.com/cs/library/usermanuals/public/XF_Cell_Mito_Stress_Test_Kit_User_Guide.pdf) accessed on 4 April 2022). The assay uses the built-in injection ports on XF sensor cartridges to add modulators of respiration into cell well during the assay to reveal the key parameters of mitochondrial function. The modulators included in this assay kit are Oligomycin, Carbonyl cyanide-4 (trifluoromethoxy) phenylhydrazone (FCCP), Rotenone, and Antimycin. This figure illustrates the injection sequence of these modulators and the parameters that are obtained with this assay.

### 2.2. Animal Studies

All animal experiments were conducted as per Mayo Clinic Institutional Animal Care and Use Committee approval and institutional guidelines. MM1.S cells (5 × 106) were subcutaneously injected into the flank of SCID mice. Once tumors reached a median volume of ~200 mm^3^, mice were randomized to receive either vehicle (PBS, intraperitoneal injection), AT-101 (35 mg/kg per oral gavage), lenalidomide (10 mg/kg, intraperitoneal injection) and dexamethasone (0.5 mg/kg, intraperitoneal injection) or the combination of the 3 drugs (ARd), daily for 10 consecutive days. Orthotopic tumor volume was assessed by caliper measurements and quantitated by using the formula: 4/3πabc; where a, b and c represent length, width and depth, respectively. Percent treatment/control (T/C%) values were calculated using the formula:(1)$$\frac{T}{C}\%=100\times \frac{\Delta T}{\Delta C}$$

Systemic disease burden was also measure by ELISA quantifying human Ig-lambda (Hu-Igλ), which is secreted by MM1.S cells and isolated from blood sera collected from mice.

This open label, single institute, Phase I study enrolled patients with relapsed/refractory MM. Patients with relapsed/refractory MM who required treatment as per standard guidelines [24] were enrolled. The primary objective was to determine the safety and efficacy of escalating doses of AT-101 in combination with a fixed dose of Rd on a 4-week treatment cycle. Each treatment cycle spanned 28 days with AT-101 administered once daily on days 1–21. AT-101 dosing was designed to reach a maximum daily target of 20 mg (Cohort 1; 10 mg oral daily on days 1–21, Cohort 2; 20 mg oral daily on days 1–21) utilizing a standard 3 + 3 dose escalation design. For cycle 1, patients received AT-101 alone; the rationale for which was to isolate its potential clinical activity as a single agent and carry out correlative studies. From Cycle 2 onwards, AT-101 was given in combination with lenalidomide 25 mg oral daily on days 1–21 and dexamethasone 40 mg oral on days, 1, 8 and 15. Treatment was given for a fixed duration of 12 months/cycles and patients were followed thereafter for progression. Secondary objectives included toxicities, overall response rate (ORR), and progression free survival (PFS) associated with the ARd regimen. Clinical responses and disease progression were assessed by the investigator, using the International Myeloma Working Group criteria [24]. Protein complex electrochemiluminescence ELISA and immune cell analysis correlative studies were conducted on primary biospecimens. These studies were conducted according to Good Clinical Practice guidelines in accordance with the Declaration of Helsinki, and under the approval of the Mayo Clinic institutional review board approval. The distribution of progression-free survival was estimated using the method of Kaplan and Meier. Statistical analysis was performed by SAS 9.4 biostatistical software (SAS Institute, Cary, NC, USA).

Additional details on the translational experiments and clinical trial are provided as Appendix A.

## 3. Results

### 3.1. AT-101 Reduces MM Cell Viability and Leads to Mitochondrial-Mediated Apoptosis

The Bcl-2 family consists of multiple anti-apoptotic proteins that bind and block the function of pro-apoptotic family members. Among these, the Bcl-2 and Mcl-1 are known to safeguard MM cell survival [25]. Indeed, analysis of 29 MM cell lines uncovered that *MCL1* gene expression is significantly higher than that of *BCL2* (Appendix A.) We therefore examined the relative essentiality of *BCL2* and *MCL1* towards MM cell survival by analyzing data from several CRISPR/Cas9 loss-of-function genetic screens in the CancerDep Map database. While loss of *BCL2* was noted to be essential in nearly all 29 MM cell lines (median CERES score −0.2), loss of *MCL1* appeared more detrimental to MM cell viability (median CERES score −1.0) (Appendix A). These results are clinically relevant as MM patients who have higher *MCL1* gene expression have shorter progression free survival (PFS) (Appendix A). Mcl-1 is implicated in resistance towards ABT-199 in MM and so we reasoned that dual-inhibition of Bcl-2 and Mcl-1 would be more lethal to MM cells vs. inhibition of Bcl-2 or Mcl-1 alone. Unfortunately, the use of highly specific Mcl-1 inhibitors, have been shown to be associated with increased cardiac toxicity in MM patients [26]. Thus, we hypothesized that use of a single drug that inhibits both Bcl-2 and Mcl-1; yet carries a favorable safety profile could achieve our desired effect. We have previously reported that AT-101 targets Bcl-2 and Mcl-1 and induces cell death in malignant B-lymphoid cancers, including MM [21]. Indeed, NMR data (generated by the Wang lab at University of Michigan) [27] as well as fluorescence polarization assays carried out by several others [28,29,30,31] show AT-101 binds to Bcl-2 family members within the BH3 groove (Appendix A). Expanding on our prior findings, AT-101 more effectively reduced cell viability in KMS-11 and RPMI-8226 cells with a mean IC_50_ of 2.45 uM compared to ABT-199 (mean IC_50_, 8.9 uM, Figure 1A). Loss of cell viability upon exposure to AT-101 was due to apoptosis and significant in both MM cell lines as well as primary CD138+ tumor cells from MM patients (Figure 1B–D).

AT-101 induces cell death via compromising critical mitochondrial function [21,32]. Indeed, we noted that treatment of MM cell lines with AT-101 led to an increase in outer mitochondrial membrane permeability (Figure 1E,F), which indicates impending loss in the electron transport chain (ETC). Mitochondrial bioenergetics analysis (Figure 1G) showed complete mitigation of the ETC, significant compromise in basal respiration, maximal respiration and loss of ATP production. To confirm mitochondrial ATP blockade, we performed an orthogonal assay and noted the same trend in both KMS-11 and RPMI-8226 cells (Appendix A). Further, RNA-Seq analysis showed modulation of several genes involved in the S-phase of the cell cycle and those implicated in mitochondrial function in AT-101 vs., vehicle or ABT-199 MM cells Figure 1H,I, Appendix A).

### 3.2. AT-101 Enhances the Anti-Tumor Activity of Rd in a Xenograft Model of MM

We have previously shown that AT-101 boosts the cytotoxic activity of lenalidomide alone in preclinical models of CLL, MM and WM [21]. Mechanistically, we uncovered that AT-101 + lenalidomide and dexamethasone (ARd) upregulated several tumor-suppressor genes in MM/WM cell lines treated with the combination; central to which was *IRF1*. *IRF1* targets several downstream pathways, including ERK signaling to block neoplastic proliferation. Examining this further here, we noted a marked decrease in both ERK and phospho-ERK expression in ARd-treated MM cells (Figure 2A), which corroborated our prior data probing the mechanisms underlying the in vitro cytotoxicity of the ARd combination. Notably, no change in cereblon (target of lenalidomide) was noted with the triple drug combination (Appendix A).

To validate the ability of AT-101 to augment activity of Rd in vivo, we tested the tumor growth-inhibitory potential of the combination in a xenograft model of MM (Figure 2B). Expectedly, Rd treatment reduced tumor growth (T/C, 84%) in mice, which was not significantly different from that observed in mice treated with AT-101 alone (T/C, 80.1%). However, in mice treated with the ARD combination, treatment resulted in a more significant reduction in tumor burden (T/C, 64.8%, *p* < 0.001) (Figure 2C). In addition to local/orthotopic disease burden, we also measured secretion of human lambda light chains (Hu-IgL) secreted by the xenografted MM1.S cells as a marker of systemic disease burden. Median Hu-IgL levels reached 242 ng/mL and 130 ng/mL by Day 24 in control and Rd-treated mice, respectively. These levels did not exceed over 105 ng/mL in ARd-treated mice–a difference that was statistically significant (Figure 2D). Overall, the ARd combination was well-tolerated and no significant weight loss that would indicate acute toxicity from either AT-101, Rd or ARd were observed (Figure 2E). These data coupled with that previously published by us and others on AT-101 [33] provided for the rationale for clinical testing of the ARd regimen.

### 3.3. Clinical Validation: Patient Demographics and Clinical Characteristics

The primary goal of this phase I study was to establish safety and define the toxicity profile of the ARd regimen. A total of ten patients were enrolled in this abbreviated 2-cohort, phase I portion of the study. Detailed demographic information of the patients enrolled is provided in Table 1. Briefly, the median age of the patients enrolled was 68.5 years (range 55–75), the median time since initial diagnosis of MM was 4.5 years (range 0.6–8.3). Most patients had either advance stage disease (ISS II/III in 70%) or high-risk cytogenetics (80%). Patients had received a median of 2 prior lines of therapy (range 1–3). Patients had been previously treated with regimens containing lenalidomide (100%), bortezomib (90%), dexamethasone (100%), daratumumab (30%) and high-dose chemotherapy and autologous stem cell transplant (70%). Among these, 3 (30%) patients were lenalidomide-refractory [34] while 2 (20%) were refractory to bortezomib [34] and 3 (30%) were refractory to daratumumab. 2 (20%) patients were refractory to both bortezomib and daratumumab (Table 1). Patients were defined as being refractory to treatment if they did not achieve a minimal response to therapy, progressed while on treatment or developed progressive disease (PD) within 60 days of the last treatment dose [34].

## 4. Disposition

The median duration of treatment was 7.5 cycles (range 2–12) and 3 patients completed all planned 12 cycles of treatment. The primary reason for discontinuation was disease progression. 5 patients developed PD while on treatment, 2 patient developed stable disease but were removed from the trial due to an adverse event (1 patient due to immunosuppressed state and 1 patient due to anemia). 4 patients received 10 mg of AT-101 and 6 patients received 20 mg of AT-101. There were no dose reductions of AT-101. However, 2 patients on dose level 2 (20 mg of AT-101) had dosing errors. One patient was given 10 mg for the first 7 days of Cycle 1 due to a pharmacy error, after which the dose was corrected to 20 mg. The second patient took 10 mg the entire first cycle due to patient error. Six patients had dose reductions in lenalidomide on 9 cycles due to hospitalization and neutropenia (1 patient); neuropathy (2 patients); neutropenia and cytopenia (1 patient); elevated creatine clearance (1 patient); fatigue and anemia (1 patient). Four patients required dose reductions in dexamethasone on 5 cycles due to anemia and fatigue (1 patient), nausea and dizziness (1 patient), fatigue & irritability (1 patient), and intolerability (1 patient). Four patients did not require any dose reductions.

## 5. Safety Profile

The most common AEs were hematologic toxicities (thrombocytopenia [90%], anemia [90%], low white blood cell count [80%], neutropenia [90%]) and gastrointestinal toxicities (diarrhea [50%], constipation [30%], and nausea [30%]. The most common grade 3/4 adverse events were cytopenias; thrombocytopenia (20%), low white blood cells, (30%), anemia, (30%) and neutropenia (50%). G3/4 adverse events (AEs) included atrial flutter (*n* = 1), neutropenia (*n* = 5), febrile neutropenia (*n* = 1) and thrombocytopenia (*n* = 2), back pain (*n* = 1), and white blood cell reduction (*n* = 3). Any grade non-hematologic AEs seen in at least 20% (*n* = 2) patients included fatigue (*n* = 10), peripheral sensory neuropathy (*n* = 6), nausea (*n* = 3) diarrhea (*n* = 5), constipation (*n* = 3), and creatinine increased (*n* = 2) (Table 2). No events of laboratory or clinical tumor lysis syndrome were reported in the study.

Nine patients were evaluable for MTD determination (3 patients dose level 1, 6 patients dose level 2). The dose level 2 patient who took 10 mg AT-101 the entire first cycle due to patient error was considered not evaluable and was replaced. DLTs at 20 mg daily dose of AT-101 with 25 mg of R and 40 mg weekly (dose level 2) included one patient with grade 4 febrile neutropenia and grade 4 neutropenia lasting 9 days and one patient with grade 4 thrombocytopenia lasting 8 days. Since 2/6 patients at dose level 2 had a DLT and only 3 patients were evaluable for MTD determination at dose level 1, the MTD was not determined.

## 6. Efficacy

Ten patients were included in the efficacy analysis. After one cycle of single agent AT-101, 9 patients had stable disease and one patient had unconfirmed disease progression. For patients who received at least >1 full cycle of the ARd triple drug regimen, The ORR was 40% (2 each with very good partial response and partial response) and clinical benefit rate (CBR) was 90% with 2 additional patients showing minor response and 3 experiencing stable disease (SD) (Figure 3A). One patient experienced progressive disease after 2 cycles of treatment. In patients with high-risk disease, the ORR was 25% and the CBR was 83.5%.

Overall, two patients have died, and 6 patients have progressed. Both deaths were due to disease progression. Median follow-up in patients still alive was 16.5 months (range: 7.0–24.0). The median number of cycles completed before development of progressive disease was 7.0 (range: 2–12). The median PFS for all patients was 14.9 months (95% CI 7.1-NR) (Figure 3B). The median PFS for patients with HR cytogenetics was not reached (NR) (95% CI 7.1 months -NR).

## 7. Correlative Analyses

As AT-101 disrupts the interactions between Bcl-2 and BIM and separately between Mcl-1 and Noxa, we measured Bcl-2:Bim and Mcl-1:Noxa complex levels in MM cells from patients treated with ARd. Relative to pre-treatment levels, Bcl-2:Bim complexes were significantly lower in post-treatment tumor cells from patients 1 and 5 (Figure 4A). Conversely, a significant increase in Mcl-1:Noxa complexes was noted in these same patients. In patients 3 and 7, Bcl-2:Bim complex formation was found to increase, however, Mcl-1:Noxa complexes were significantly decreased (Figure 4B). These observations suggested on-target activity of AT-101 on the disease driving anti-apoptotic protein- as in the case of Pts. 1 and 5 (decrease in Bcl-2:Bim complexes) and which was potentially compensated for an increase in Mcl-1:Noxa levels by the tumor cells to maintain their long-term survival.

Next, we examined for changes in immune cell fractions in Pts. 3, 7 and 8 which we hypothesized would be evident after treatment with ARd. Relative to pre-treatment, we found a statistically significant increase in Th-effector cells, cytotoxic CD8+ T-cells and NK cells in post-treatment BM aspirate (Figure 5A–C). Contrastingly, a significant decrease in immunosuppressive T-regulatory and B-regulatory cells was noted after 1 complete cycle of the combination therapy (Figure 5D,E). These findings highlight the ability of the ARd drug regimen to promote changes in the tumor microenvironment, which restore anti-tumor immunity. Clinical characteristics of these patients is described in Appendix A.

## 8. Conclusions

This is the first reported clinical trial combining a Bcl-2 + Mcl-1 targeting drug with an IMiD in MM. Our results show the combination of AT-101 and Rd is a clinically active regimen which has an acceptable toxicity profile and an ORR of 40% in a predominantly high-risk, relapsed/refractory MM patient population

MM cell survival has been shown to be dependent on Bcl-2, Mcl-1 or both [9]. This biological plasticity in usage of one or both anti-apoptotic proteins by MM cells presents a unique challenge in clinical development of drugs, where inhibition of the right Bcl-2 family member at the right time is the key to crippling tumor cell survival programming. Targeting Bcl-2 alone with agents such as ABT-199 has yielded limited clinical benefit in MM; except for in patients that carry the t(11;14)(q13;32)] chromosomal abnormality and whose tumor cells are more dependent on Bcl-2 compared to Mcl-1 or Bcl-xL [35]. Results from a phase II study showed that ABT-199 and dexamethasone in relapsed/refractory MM patients with t(11;14) and who had received a median of 5 prior lines of therapy, produced an ORR of 48% [36]. Although encouraging, t(11;14) is found in only 15–20% of MM patients and among those who do not harbor this genomic aberration, remission induction with ABT-199 is less than optimal (~20% ORR). This is not to say that Bcl-2 is a superfluous biological target in patients devoid of t(11;14) but rather it operates in dynamic concert with Mcl-1 or Bcl-xL. Thus, agents that collectively inhibit the supportive function of each of these important anti-apoptotic family members need to be actively sought after and developed for use in MM. In this context, other BH3 mimetics such as ABT-737 and its oral derivative navitoclax, which bind to Bcl-2, Bcl-xL and Bcl-w (but not Mcl-1) with high affinity have also been investigated in MM, but their development was precluded either due to lack of anti-tumor activity or unacceptable toxicity [37]. In this regard, the tractability of AT-101 to bind and block both Bcl-2 and Mcl-1 overcomes the limitations of the aforementioned BH3 mimetics and carries a more favorable toxicity profile. A major drawback of earlier BH3 mimetics targeting multiple Bcl-2 proteins, such as navitoclax, was their on-target/off-tumor inhibition of Bcl-xL in platelets, resulting in dose-limiting thrombocytopenia [38]. By comparison, the impact of AT-101 on platelets is less prominent, supported by its lower binding affinity for Bcl-xL (0.48 uM), compared to Bcl-2 (0.32 uM) and Mcl-1 (0.18 uM) [29]. Overall, our results show that ARd therapy was well-tolerated with only 20% of patients developing grade 3/4 thrombocytopenia; more than likely attributable to lenalidomide (where this rate is ~15%)[39] and which improved after dose reduction. Notably, none of these patients required dose reductions in AT-101 and none were taken off the study completely due to occurrence of adverse effects.

Therapeutic regimens incorporating drugs that target Bcl-2 proteins must be carefully designed with both biological mechanisms and clinical feasibility in mind. Approaches combining PI with BH3 mimetics have been investigated based on preclinical studies, which showed enhanced apoptosis of MM cells, upregulation of NOXA and downregulation of Bcl-xL. While complementary from a biomechanistic standpoint, this combination presented significant clinical challenges. In the phase III BELINI trial comparing bortezomib, ABT-199 and dexamethasone to bortezomib, dexamethasone and placebo, a clear benefit in median PFS was noted in the ABT-199 group (22.4 months) versus the placebo group (11.5 months, HR 0.63; *p* = 0.010) [18]. As anticipated, in 35 patients with t(11;14), the median PFS was not reached with ABT-199 and was 9.5 months with placebo (HR 0.11; *p* = 0·0040) [18]. Despite these encouraging results, an increased proportion of overall survival events in the intention-to-treat population was observed in the ABT-199 group; with a 2-fold increase in the numbers of deaths occurring in the ABT-199 group compared with the placebo group (21% vs. 11%, HR 2.03; *p* = 0.034) [18]. This increased death signal was not noted in patients with t(11;14) and overall suggest that ABT-199-based regimens are best suited for patients with t(11;14).

Considering these results, our rationale for the ARd regimen was strategically devised accounting for MM cell biology, disease immunology and clinical feasibility in MM patients (who may or may not harbor t(11;14)). Gene expression profiling of MM cells treated with ARd results in upregulation of *IRF1* [21], which is a tumor suppressor that inhibits *BCL2* transcription [40], but increases expression of caspases and enhancement of mitochondrial-mediated tumor cell apoptosis. Indeed, this functional effect in activation of caspases and apoptotic machinery appears to be preserved even when AT-101 and dexamethasone are combined with the second-generation IMiD, pomalidomide (Appendix A).

Pharmacodynamic studies carried out by us attest to the biological effects of ARd therapy in MM patients. In tumor cells isolated from patients on the trial, we noted a significant decrease in Bcl-2:Bim or Mcl-1:Noxa complexes after 1 full cycle of treatment, suggesting increased availability of Bim to trigger mitochondrial-mediated programmed cell death. Bcl-2 family members are highly dynamic and can shift reliance from one anti-apoptotic partner to another in response to therapy-induced stress. We believe that in Pts. 3 and 7, Mcl-1:NOXA interactions were more important in preserving cellular survival as compared to Bcl-2:BIM interactions, whereas the opposite may be true for patients 1 and 5. Indeed, further examination is warranted in additional samples. Activation of tumor apoptotic machinery (instigated by AT-101) potentially leads to increased leakage of pro-inflammatory cytokines from the tumor cells, which serve as a chemoattractant for neighboring anti-tumor immune-effector cells (stimulated by Rd) and promote immunogenic cell death. Our hypothesis is supported by correlative studies showing that MM patients responsive to ARd had significantly higher CD4+ effector, CD8+ cytotoxic and NK cells but lower immunosuppressive T-regulatory and B-regulatory cells in post-treatment versus pre-treatment BM samples. Thus, biologically, the combination AT-101 and Rd taps into tumor-intrinsic and extrinsic mechanisms, respectively, that are complementary but mutually exclusive of each other and culminate in enhanced disease control. Therefore, clinically, ARd represents an all-oral therapeutic regimen in which the partnering drugs carry no overlapping toxicity and whose AE profile should be easily manageable by most physicians familiar with administering Rd.

Although our results are based on a small cohort of patients, the ARd combination was noted to be active, patient convenient (due to all an oral regimen) and prolonged the PFS of patients who had previously received a PI, anti-CD38 targeting mAb, chemotherapy and/or ASCT. Correlative analysis suggests the combination to support anti-tumor immunity which is key to long-term disease control. Overall, our clinical results are encouraging, and support continued investigation of this novel drug combination.

## Figures and Tables

**Figure 1 cancers-15-00477-f001:**
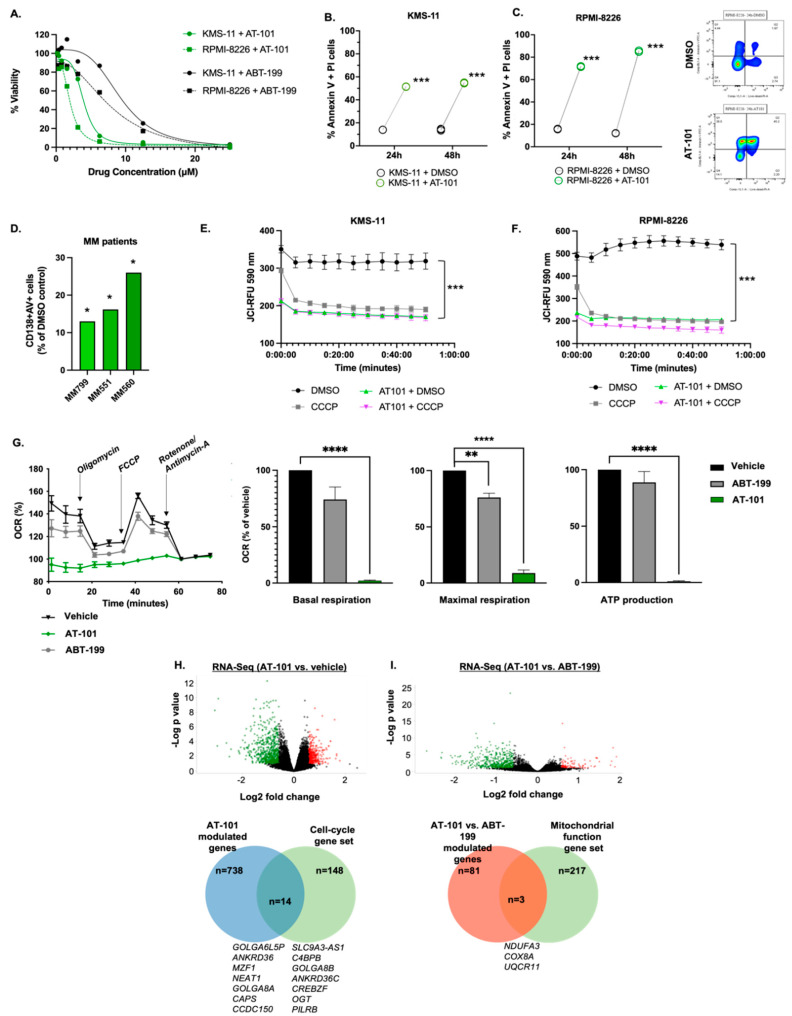
AT-101 induces myeloma cell death and compromises mitochondrial functions. (**A**) Multiple myeloma (MM) cell lines were treated with increasing concentrations of AT-101 or ABT-199 for 72 h and IC50 for each drug was determined by CellTiter Glo 2.0 assay. The IC50 of AT-101 in KMS-11 and RPMI-8226 cell was 3.9 uM and 2.0 uM, respectively. The IC50 of ABT-199 in these cells was 9.3 uM and 8.1 uM, respectively, which was significantly higher than that of AT-101 (*p* = 0.002). (**B**,**C**) MM cell lines were treated with DMSO (vehicle) or AT-101 (5 uM) for 24 h and 48 h and cell death was determined by staining with annexin-V + PI followed by flow cytometry analysis. A representative contour plot displaying cell death in RPMI-8226 cells is shown in the panel on the right. (**D**) Apoptosis was assessed in bone marrow mononuclear cells from MM patients treated ex vivo with AT-101 or DMSO for 24 h, followed by labeling with an anti-CD138 antibody, annexin-V and analysis by flow cytometry. (**E**,**F**) Mitochondrial outer membrane permeability was measured in KMS-11 and RPMI-8226 cells treated with AT-101 (5 uM, 24 h), which showed a significant increase in permeability (decrease in JC1 dye relative fluorescence units [RFU]) measured over a 1 h period. The uncoupling agent CCCP was used as an assay positive control. (**G**) Mitochondrial respiration and bioenergetics was measured in KMS-11 cells treated with AT-101 (5 uM, 24 h) or ABT-199 (5 uM, 24 h) and from which basal respiration, maximal respiration and mitochondrial ATP production was calculated. RNA-seq was performed to determine changes in transcriptome of KMS-11 cells exposed to AT-101 or ABT-199 (1 uM, 24 h). Volcano plots show genes differentially and significantly expressed between cells treated with (**H**). AT-101 (vs. DMSO) and (**I**). AT-101 vs. ABT-199. Pathway analysis was performed and identified 14 genes within the Cell cycle geneset that were differentially altered in AT-101 vs. DMSO cells (*p* < 0.001) and 3 genes within the Mitochondrial function geneset differentially expressed between AT-101 vs. ABT-199 (*p* < 0.001). Pre-/Post- plots show individual data points (each point average from a single experiment). Bar graphs show data presented as mean ± SD. Mitochondrial membrane potential Each experiment was performed three times in triplicate, except for the mitochondrial bioenergetics assay (conducted three times with 6 replicates) and those performed on primary MM samples. For apoptosis assessment in MM patient CD138+ cells, DMSO-treated samples were grouped together as “Control” and AT-101-treated samples were group together as “Treatment” and unpaired T test was applied (type 2 = two sample equal variance-homoscedastic). (* *p* < 0.05; ** *p* < 0.001, *** *p* < 0.001, **** *p* < 0.0001).

**Figure 2 cancers-15-00477-f002:**
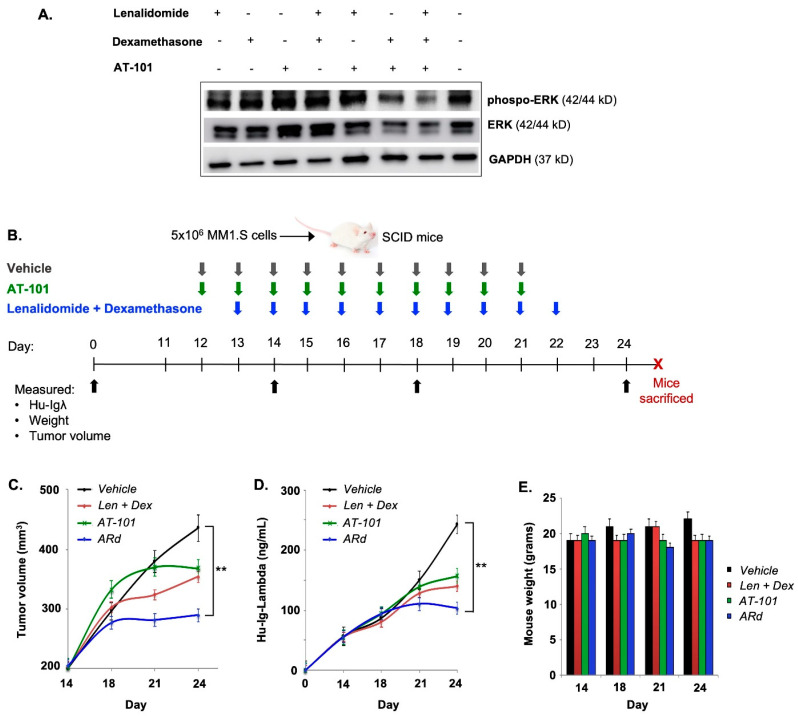
AT-101 enhances the in vivo anti-tumor effects of lenalidomide and dexamethasone in a xenograft model of human myeloma. (**A**) Western blot analysis for ERK and phospho-ERK (pERK) in KMS-11 cells treated with lenalidomide (1 uM), dexamethasone (1 uM), AT-101 (1 uM) or the combination (ARd) for 24 h relative to untreated cells. (**B**) In vivo activity of AT-101 in combination with lenalidomide and dexamethasone Female SCID mice were xenografted with human lambda light chain (Hu-Igλ) secreting MM1.S myeloma cells. Once tumors reached a median volume of ~200 mm^3^ (Day 14), mice were randomized to 4 groups (*n* = 10/group) receiving either DMSO (vehicle), AT-101 (35 mg/kg per oral gavage), lenalidomide (10 mg/kg, intraperitoneal injection) and dexamethasone (0.5 mg/kg, intraperitoneal injection) or the combination of the 3 drugs (ARd), for 10 consecutive days. (**C**) Tumor volume was estimated every 4 days post-randomization and reached an average of ~410 mm^3^ in the control group by Day 24. This contrasted with the mice that received treatment with Ld or AT-101, whose tumors grew to an average of ~330 mm^3^. A significant reduction in tumor volume was observed in mice treated with the ARd combination, with most mice showing little to no increase in tumor size beyond 254 mm^3^. (**D**) Next, we examined Hu-Igλ secretion in mice sera by ELISA and consistent with tumor growth pattern, we observed significantly lower Hu-Igλ levels in mice treated with LDA. (**E**) Notably, treatment with either AT-101, lenalidomide and dexamethasone or ARd did not result in significant weight loss or other toxicities. Data shown are mean ± SD (** *p* < 0.01). Uncropped WB images were shown in Appendix A.

**Figure 3 cancers-15-00477-f003:**
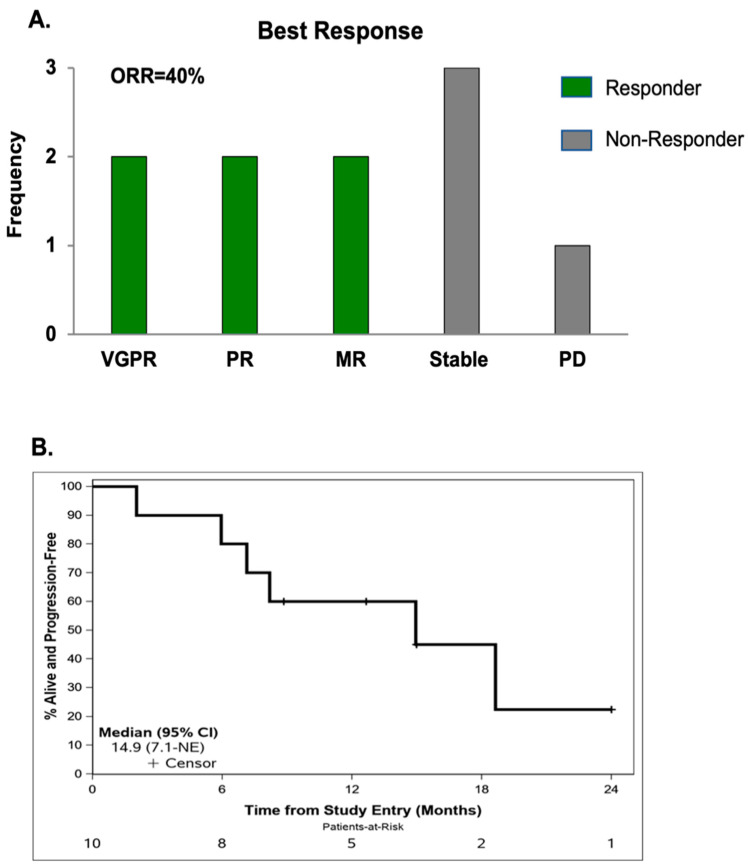
Clinical response and progression free survival (PFS) in MM patients treated with the ARd regimen. (**A**) Overall response rate (ORR) was assessed in 10 patients treated with ARd and was 40% (2 VGPR, 2 PR, 2 MR, 3 SD, 1 PD). (**B**) The median PFS for all patients was 14.9 months (95% CI 7.1-NR).

**Figure 4 cancers-15-00477-f004:**
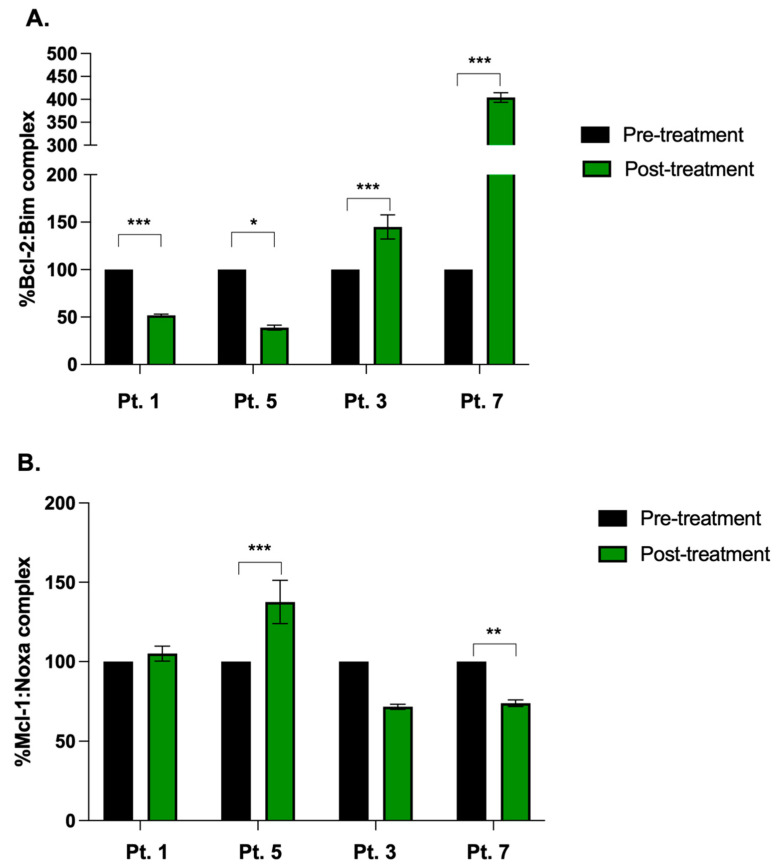
Binding interaction between anti-apoptotic and pro-apoptotic Bcl-2 family proteins in patients on ARd treatment. MSD-ELISA was performed on MM tumor cells isolated from 4 MM patients at baseline (pre-treatment) and after 2 cycles of ARd (post-treatment) showing (**A**) % of Bcl-2:Bim complex formation or (**B**) % of Mcl-1:Noxa complex formation. Data shown are mean ± standard error of the mean (* *p* < 0.05; ** *p* < 0.001, *** *p* < 0.001).

**Figure 5 cancers-15-00477-f005:**
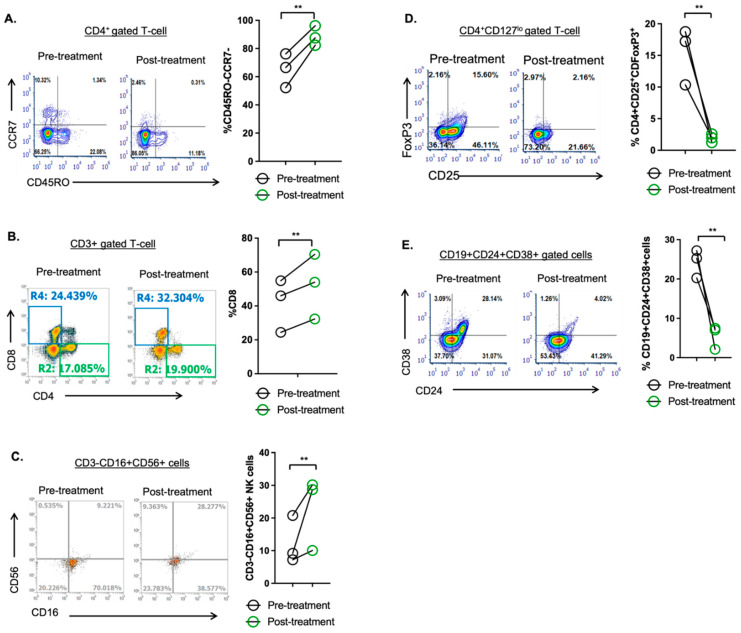
ARd therapy promotes anti-tumor immune reconstitution in MM patients. Flow cytometry was performed on bone marrow mononuclear cells isolated from MM patients who achieved >PR. Relative to pre-treatment, (**A**) CD4+ Th effector cells, (**B**) CD8+ cytotoxic T-cells and (**C**) NK cells were significantly increased in post-treatment samples whereas pro-tumor (**D**) T-regulatory and (**E**) B-regulatory cells were significantly decreased. Each individual data point represents a single patients’ data. ** *p* < 0.001.

**Table 1 cancers-15-00477-t001:** Patient demographics and clinical characteristics (*n* = 10).

Sex	
Female	4 (40%)
Male	6 (60%)
Age, median (range), y	69 (56–74)
Time from diagnosis to trial enrollment, median (range), y	4 (0–7)
**ISS stage**	
I	1 (10%)
II/III	7 (70%)
Unknown	2 (20%)
**ECOG PS**	
0	6 (60%)
1	4 (40%)
≥2	0
**Cytogenetic Abnormalities**	
High Risk Cytogenetics	8 (80%)
Del(17p)	4 (40%)
1q dup	4 (40%)
T(11;14)	1 (10%)
**Induction Regimen**	
VRD	8 (80%)
RD	1 (10%)
CyBorD	1 (10%)
Autologous Stem Cell Transplant	7 (70%)
Lines of Prior Therapy, median (range)	2 (1–2)
Lenalidomide Refractory	3 (30%)
Bortezomib Refractory	2 (20%)
Daratumumab Refractory	(30%)

VRD; bortezomib-lenalidomide-dexamethasone; RD; lenalidomide-dexamethasone; CyBorD; cyclophosphamide-bortezomib-dexamethasone.

**Table 2 cancers-15-00477-t002:** Treatment-Emergent Adverse Events (*n* = 10).

AE	Any Grade	Grade 3/4
Any AE	10 (100%)	8 (80%)
**Hematologic Events**		
Thrombocytopenia	9 (90%)	2 (20%)
Low White Blood Cell Count	8 (80%)	3 (30%)
Anemia	9 (90%)	3 (30%)
Neutropenia	9 (90%)	5 (50%)
**Gastrointestinal Events**		
Diarrhea	5 (50%)	0
Constipation	3 (30%)	0
Nausea	3 (30%)	0
Vomiting	1 (10%)	0
**All other AEs**		
Fatigue	10 (100%)	0
Back Pain	1 (10%)	1 (10%)
Peripheral Neuropathy	6 (60%)	0
Hyperglycemia	3 (30%)	0
Hyperkalemia	2 (20%)	0
Hypokalemia	6 (60%)	1 (10%)
Hyponatremia	4 (40%)	0
Atrial Flutter	1 (10%)	1 (10%)
Febrile Neutropenia	1 (10%)	1 (10%)
Increased Creatinine	2 (20%)	0

## Data Availability

The data presented in this study are available on request from the corresponding author. The data are not publicly available due to patient privacy.

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
