# Peer review of "AT-101 Enhances the Antitumor Activity of Lenalidomide in Patients with Multiple Myeloma"

_cancers, 2023, doi:10.3390/cancers15020477_

Round 1

Reviewer 1 Report (Previous Reviewer 1)

The authors of the manuscript have not responded to the objections I raised in the review of the first version.

As I said, many data in the literature indicate that AT-101 (gossypol) is not a Bcl-2 inhibitor in cell culture since it induces apoptosis in the absence of Bax and Bak.

AT-101's mechanism of action is therefore different from other Bcl-2 inhibitors, such as ABT-199. It is true that ABT-199 has metabolic effects that are independent of Bcl-2, but its mechanism of induction of apoptosis is Bcl-2 inhibition.

Thus, in my opinion, all the data and description of AT-101 as a Bcl-2/Mcl-1 inhibitor should be deleted and the text should be changed describing AT-101 as a proaoptotic compound but not a Bcl-2/Mcl-1 inhibitor.

Author Response

Reviewer 1 suggests removing the statement that AT-101 induces MM cell death through Bcl-2 inhibition due to evidence in the literature showing that gossypol caused cell death in BAK/BAK KO cells. However, some of the references being cited by Reviewer 1 do not actually fully support his argument. For example, his reference to Vogler et al Cell Death Differ. 2009, it actually states that “cell death induction by gossypol and apogossypol are in fact partially due to their Bcl-2-inhibitory function” https://www.nature.com/articles/cdd200948. Indeed, AT-101 may trigger alternative death signaling pathways inside the cell (no different from ABT-199 or any other drug for that matter), but it does not deny that it binds to Bcl-2 or Mcl-1 within the BH3 groove in the same physical space as pro-apoptotic BH3 family members (see Zhai et al Cell Death Differ. 2006; Wang et al J. Med. Chem. 2006; R.M. Mohammad et al AACR, 2006, Wei et al J. Med. Chem, 2009).

Furthermore, we found that there is a BAK/BAX independent but BCL2-related cell death signaling pathway (Lei et al The FASEB Journal, 20: 2147-2149) triggered by gossypol or AT-101. Indeed these investigators demonstrated that gossypol binding to Bcl-2 may cause a conformational change in the structure of Bcl-2 allowing for it directly breach mitochondrial integrity and activate apoptosis that is independent of mitochondrial transition pores by Bax/Bak. https://faseb.onlinelibrary.wiley.com/doi/epdf/10.1096/fj.05-5665fje . This further suggests that BAX/BAK KO cells cannot be conclusively used as a 100%-faithful model to exclude AT-101 from being labelled as binding and targeting Bcl-2 and its sister proteins that contain a BH3 domain. 

Reviewer 2 Report (Previous Reviewer 2)

I am staisfied with the revisions.

Author Response

Thank you for your comments. We have made all the revisions requested. 

Reviewer 3 Report (New Reviewer)

The manuscript by Ailawadhi et al. reports the ability of AT-101 to bind and inhibit Bcl-2 and Mcl-1 in vitro. Authors also present an evaluation of AT-101, lenalidomide and dexamethasone treatment in xenograft mouse model and a clinical proof of concept of safety and efficacy of the regimen in relapsed/refractory multiple myeloma patients.

Despite being a potentially interesting approach for MM treatment by targeting both Bcl-2 and Mcl.1, the soundness of the results is somehow occluded by the shortness of the methods section. Methodology is severely undescribed and this needs to be fixed prior publication since several aspects of the sections 3.1-3.3 in results can hardly be evaluated. Descriptions in the figure legends do not provide a detailed idea of the methods employed, nor the materials.

Other minor aspects:

1)    References appear systematically at the beginning of the following sentence instead of being at the end of the corresponding phrase. Please check throughout the manuscript.

2)     Ident formatting from section 3.2 needs to be revised.

3)    Line 363. “Please clarify the meaning of MM; except for in patients harbouring”.

4)    Affinity constants units are shown as “uM” in figure legends and throughout the manuscript. I presume this should be mM instead ?.

Author Response

We have removed Section 3.2 and inserted as a Supplemental section/Fig. 3. We have inserted full methodology for the animal experiment and the Seahorse assay. We have also elaborated a bit more in the figure legends. All other experiments are quite standard, and their full description / methodology is and has always been in the Supplemental Materials & Methods document.

Other minor aspects:

1)    References appear systematically at the beginning of the following sentence instead of being at the end of the corresponding phrase. Please check throughout the manuscript. Done

2)     Ident formatting from section 3.2 needs to be revised. Done

3)    Line 363. “Please clarify the meaning of MM; except for in patients harbouring”.- “translocation of chromosomes 11 and 14 [t(11;14)(q13;32)], hereafter referred to as t(11;14)”. We have now restated this as: “except for in MM patients that carry the t(11;14)(q13;32)] chromosomal abnormality.”

4)    Affinity constants units are shown as “uM” in figure legends and throughout the manuscript. I presume this should be mM instead? – Please clarify in the rebuttal write-up that the units are all µM and not mM

Round 2

Reviewer 1 Report (Previous Reviewer 1)

Vogler et al.  described that the proapoptotic activity of Gossypol or Apogossypol  is decreased in Bax/BAk DKO MEFs respect WT MEFs, but these compounds induce apoptosis at higher concentration (10-30 micromolar) and this result demonstrates that these compounds induce apoptosis through a mechanism independent of Bcl-2 inhibition. One bona fide Bcl-2 inhibitor (ABT-737 in the same figure) was completely blocked even at 30 micromolar. Any proapoptotic drug that uses Bax or Bak to induce cytochrome c release by the mitochondrial pathway will be blocked in Bax/Bak deficient cells, thus gossypol is using a different mechanism to induce apoptosis.

The results described by Lei et al were very interesting, however these results were reported in 2006 and as far as I know there are no more articles describing this effect of gossypol on Bcl-2 protein. Anyway, the authors could comment this article in the discussion.

Thus, I think that in all the manuscript AT-101 has to be described as a proapoptic drug but not a Bcl-2 inhibitor. In the introduction and the discussion, the authors should describe that AT-101 induces apoptosis independently of Bax/Bak, and  they have to cite the articles that reported these results.

Author Response

Thank you for your comments.

In essence, reviewer 1 purports that AT-101 is not a true Bcl-2 inhibitor because AT-101 induces apoptosis at higher concentrations in Bax/Bak-dependent cells. However, in this manuscript we show that AT-101 binds to Bcl-2 (as shown in Supplemental Figure 3) and which has been validated by NMR / FP assay by several other groups and induces cell death via mitochondrial-mediated apoptosis (Figure 1 and Supplemental Figure 4). Furthermore, our data from clinical trial patient samples show that AT-101 decreases Bcl-2-Bim and MCl-1-NOXA complexes (Figure 4) which is proof of concept that AT-101 is a Bcl-2 and Mcl-1 inhibitor. It is possible that AT-101 has additional mechanisms of causing cell death (such as inducing ER stress) which has previously been reported in other publications (Soderquist RS, Danilov AV, Eastman A. Gossypol increases expression of the pro-apoptotic BH3-only protein NOXA through a novel mechanism involving phospholipase A2, cytoplasmic calcium, and endoplasmic reticulum stress. J Biol Chem. 2014;289:16190–9.), however we did not explore these additional mechanisms given the ample evidence shown that AT-101 does in fact inhibit Bcl-2 and Mcl-1.

We do not agree with describing AT-101 as merely a “proapoptotic drug” as we and several others before us show evidence that it is indeed a Bcl-2 and Mcl-1 inhibitor.

Reviewer 3 Report (New Reviewer)

All concerns and modifications suggested have been properlly addressed.

Author Response

Thank you for your comments. We have addressed all of the concerns and comments made by this reviewer. 

Round 3

Reviewer 1 Report (Previous Reviewer 1)

The authors have made most of the changes that I suggested.

This manuscript is a resubmission of an earlier submission. The following is a list of the peer review reports and author responses from that submission.

Round 1

Reviewer 1 Report

This manuscript describes the effect of the combination of the gossypol isomer AT-101 with lenalidomide and dexamethasone in a xenograft model of human myeloma and in a phase I clinical assay (10 patients). The first part of the manuscript analyses the effect of AT-101 in two multiple myeloma cell lines (KMS-11 and RPMI-8226) one of which (KMS-11) will be used in the xenografts.

In all the manuscript AT-101 is described and used as a Bcl-2 and Mcl-1 inhibitor. Even in Fig. 2 it is proposed a model for the binding of AT-101 to Bcl-2 and Mcl-1. Initially, gossypol was described as a Bcl-2 inhibitor, although with low affinity to Bcl-2, Bcl-X or Mcl-1 when compared with the BH3 mimetic ABT-737 (Zhai et al. Cell Death Differ, 2006). However, many data in the literature indicate that gossypol (or AT-101) is not a Bcl-2 inhibitor or BH3-mimetic. Importantly, gossypol has pro-apoptotic activity independently of Bax/Bak (Van Delft et al Cancer Cell, 2006; Vogler et al. Cell Death Differ, 2009; Villalobos-Ortiz et al. Cell Death Differ, 2020). In fact, recent data indicate that gossypol or AT-101 induce apoptosis through the induction of NOXA (Soderquist and Eastman, Mol Cancer Ther, 2016) by the activation of the unfolded protein response (Malleck and Eastman, Cancers, 2020). Thus, in my opinion all the data and description of AT-101 as a Bcl-2/Mcl-1 inhibitor should be deleted and the text should be changed describing AT-101 as a proapoptotic compound but not a Bcl-2/Mcl-1 inhibitor.

The proapoptotic efect of AT-101 has been described previously in different multiple myeloma cells lines, including RPMI-8226. Thus, for me it is difficult to understand the meaning of figure 1. Furthermore, which is the meaning of the analysis of ERk in fig 3A?

How to explain the results of Fig 5 where the Bcl-2-Bim complex after treatment was decreased in 2 patients and increased in the other two?

Reviewer 2 Report

This manuscript by Ailawadhi et al. describes a thorough and detailed description of mechanisms leading to a novel combinatorial therapy targeting Multiple Myeloma.  The study is well-conceived and the precilinical and clinical data support the authors' conclusions. I have no major criticisms, although it would have been interesting to see the dynamics of MCL-1/BIM protein complexes following treatment with ARd in responsive patients.

Minor points:

1. The title needs to be revised in a more concise style/format.

2. There are a couple of typos throughout the text which should be addressed.